

# The association between continual, year-round hunting and bellowing rate of bison bulls during the rut

Ronald J. Sarno[1], Melissa M. Grigione[2], Alessandra Higa[3], Eddie Childers[4] and Trudy Ecoffey[5]

[1] Department of Biology, Hofstra University, Hempstead, NY, United States
[2] Department of Biology, Pace University, Pleasantville, NY, United States
[3] Math and Science Department, Oglala Lakota College, Kyle, SD, United States
[4] Badlands National Park, National Park Service, Interior, SD, United States
[5] Natural Resources Conservation Service, United States Department of Agriculture, Pine Ridge, SD, United States

Corresponding author
Ronald J. Sarno,
ronald.sarno@hofstra.edu

## ABSTRACT

The impact of hunting (selective harvest, trophy hunting) on the demography of mammals is well documented. However, despite continual year-round hunting of bison in some populations, little is known about how the behavior of survivors may be altered. Therefore, in this initial study, we used focal-animal observations in adjacent populations of continually hunted and protected Plains bison (*Bison bison bison*) in western South Dakota, to examine the potential impact of hunting on bellowing rate—an important behavior that serves to intimidate rival bulls and potentially influences mate choice by females. In addition to hunting, we investigated how the number of attendant males, number of adult females, group size, and number of days from the start of rut influenced bellowing rate. Bulls bellowed an order of magnitude more often in the protected population than in the hunted populations, whereas bellowing rate was not significantly different in the hunted populations. Hunting was significantly and negatively associated with bellowing rate, while all other predictors were found to be positively associated with bellowing rate. Furthermore, the impact of hunting on bellowing rate became more pronounced (i.e., dampened bellowing rate more strongly) as the number of attendant males increased. Changes in bellowing behavior of bulls (and possibly mate choice by cows) can alter breeding opportunities. Therefore, our data suggest the need for studies with broader-scale geographical and temporal replication to determine the extent that continual year-round hunting has on bellowing rate of bison during the rut. If reduced bellowing is associated with human hunting on a larger scale, then wildlife managers may need to adjust hunting rate and duration, timing (season), and the time lag between hunting events in order to insure that bison are able to express their full repertoire of natural mating behaviors.

## INTRODUCTION

Humans have become a dominant, global evolutionary force (*Palumbi, 2001*; *Darimont et al., 2009*). Our exploitation (hunting and fishing) of wild populations has induced

rapid phenotypic and life-history changes (*Ciuti et al., 2012*; *Darimont et al., 2009*) on decadal time scales (*Darimont et al., 2009*; *Coltman et al., 2003*; *Carlson et al., 2007*). For example, in just 30 years *Coltman et al. (2003)* observed significant declines in ram weight and horn length of bighorn sheep (*Ovis canadensis*) due to unrestricted trophy hunting. Such rapid change can have profound impacts on the evolution and conservation of exploited populations because of their potential impact on population persistence (*Yoshida et al., 2003*; *Fussman, Loreau & Abrams, 2007*). While much attention has focused on how human exploitation induces phenotypic and life-history changes (*Darimont et al., 2009*; *Hendry, Farrugia & Kinnison, 2008*), there is little known about how hunting influences reproductive behavior of survivors in the numerous species of ungulates that are hunted globally (*Cromsigt et al., 2013*; *Ripple et al., 2015*).

## Management of ungulates

Many ecosystems are experiencing real (cervids in US, Europe, and Japan;) (*Côté, 2005*), or locally perceived (Asia, Tibetan wild ass (*Equus kiang*) (*Schaller, 1998*); South America, guanaco (*Lama guanicoe*) and vicuña (*Vicugna vicugna*)) (*Baldi et al., 2004*; RJ Sarno, pers. obs., 2005), increases in ungulate densities, and many of these systems require extensive management. Human hunting, either prescribed or poaching, is considered a form of predation risk that can divert time/energy from fitness-enhancing activities such as feeding, parental care, or mating (*Lima & Dill, 1990*; *Lima, 1998*). In fact, ungulates can show stronger behavioral responses to human hunting than to predation risk by large carnivores (*Proffitt et al., 2009*; *Ciuti et al., 2012*). Other potential fitness-decreasing activities associated with hunting include increased movement and frequency of changing groups in elk (*Cervus elaphus*; *Cleveland et al., 2012*; *Proffitt et al., 2009*), increased vigilance and movement in red deer (*Cervus elaphus*; *Jayakody et al., 2008*), and more frequent changes in habitat and space use in red deer, roe deer (*Cervus capreolus*), and European bison (*Bison bonasus*; *Theuerkauf & Rouys, 2008*). Flight initiation distance has also been documented to significantly increase in several hunted populations of artiodactyls (*Stankowich, 2008*).

Despite the ubiquity of harvesting ungulates for food and trophies (*Proaktor, Coulson & Milner-Gulland, 2007*; *Ripple et al., 2015*), an understanding of the population-wide effects of hunting on the behavior of survivors remains fragmentary. Events commonly associated with hunting such as firearm discharge, observing downed conspecifics, pursuit by motor vehicles, and/or the presence of humans in close proximity for extended periods of time may produce behavioral changes in survivors. Associations with any of these possibly traumatic events can lead to sensitization to human presence, which in turn, may increase vigilance or stress levels, or suppress functional behaviors such as grooming, feeding, or vocalizations (*MacArthur, Geist & Johnston, 1982*). To our knowledge, the potential impact of hunting on behaviors explicitly tied to male reproductive ecology (e.g., mate advertisement, threat displays to potential competitors, fighting) has not been investigated.

## Ecology and management of bison

This gap in the literature is particularly important because population-level responses to human activities will be crucial to successful restoration and long-term conservation of the

iconic plains bison (*Bison bison bison*). Initial observations indicated that bulls in hunted populations were less vocal than those in a protected population (RJ Sarno, pers. obs.). Therefore, we hypothesized that in a risky environment (i.e., human hunting), the best strategy for reproductively-active bulls may be to dampen vocalizations, because bellowing bulls may be targeted by hunters' sooner than other animals. Bison form large aggregations during the breeding season (rut) when mature bulls join mixed-sex and age groups. Males exhibit a linear dominance hierarchy, with older, larger bulls dominant to smaller, younger bulls (*Komers, Messier & Gates, 1994*; *Roden, Vervaecke & Van Elsacker, 2005*). Dominant males temporarily consort with cows prior to or during estrus and attempt to keep all other bulls away by engaging in vocalizations, threat displays, and fights.

The most conspicuous and frequent vocalizations made by bison bulls during the rut are bellows. Bellows function to intimidate rival males (*Berger & Cunningham, 1991*; *Berger & Cunningham, 1994*), while bellow amplitude has been linked to male body condition (*Wyman et al., 2008*). Importantly, amplitude and bellow quality are directly tied to mating success (*Wyman et al., 2008*). Vocalizations by males of different species also provide information on female reproductive status (*Semple & McComb, 2000*), influence female mate choice (*McComb, 1991*; *Reby et al., 2010*), accelerate estrus (*McComb, 1987*), and deter rival males (*Clutton-Brock & Albon, 1979*; *Bowyer & Kitchen, 1987*) and predators (*Tilson & Norton, 1981*). Therefore, any changes in bellowing behavior due to perceived predation risk (*Lima & Dill, 1990*; *Lima, 1998*) could have profound consequences for bison reproductive ecology (*Berger & Cunningham, 1994*; *Wyman et al., 2008*).

Given the importance of bellows during the mating period, and the use of hunting as a management tool, it is essential that we understand the degree to which hunting, or associated events, influence mating behavior of survivors. Therefore, our primary objective was to use focal-animal observations to compare bellowing frequency of male bison in three populations during the rut (one protected and two hunted) in western South Dakota. In addition, we assessed the influence of possible predictor variables on the frequency of bellowing rate. We hypothesized that if continual, year-round hunting of bison alters bellowing rate, then bulls in hunted populations will bellow less frequently than bulls in protected populations during the rut.

## METHODS

### Study Area

#### *Badlands National Park (protected population)*

Badlands National Park (43.8554°N, 102.3397°W; 194.3 km$^2$) includes a population of bison not subject to annual hunting mortality and is located approximately 113 km east of Rapid City, and 13 km south of Wall, South Dakota, USA. We conducted behavioral observations in the Sage Creek Wilderness Area, located in the Northern Unit of the park. The study area varies from 800 to 1,100 m in altitude and the landscape is characterized as relatively flat, open grassland with intermittent rolling hills, draws, mesas, and buttes (*National Park Service, 2003*). The dominant grassland species is western wheatgrass (*Agropyron smithin*). Wheatgrass/green needlegrass (*Nasella viridula*) associations occur

on small hills, slopes, and buttes. Rocky Mountain juniper forests (*Juniperus scopulorum*) are the most common forest type and dominate the drier slopes, butte edges, and upper draws of the study area (*National Park Service, 2003*).

Badlands National Park is characterized by hot, dry summers, and cold, dry winters. Weather is variable, and temperature extremes range between −40°–47 °C. Average yearly precipitation is nearly 40 cm, and most occurs between April and September. Average daily maximum temperature during the warmest month (July) is 33 °C (BNP National Weather Service record of river and climatological observations).

### Pine ridge reservation (two hunted populations)

The Pine Ridge Indian Reservation is a reservation of the Oglala Sioux in South Dakota, USA, and was established in 1889 in the southwest corner of the state. Pine Ridge Reservation (43.2731°N, 102.7445°W) bison pastures (hunted populations) are located in the central and southwestern portion of the reservation in Bennett County, South Dakota. Pastures range in elevation from 762 to 1,219 m. Annual precipitation varies from 33 to 48 cm, and summer/winter temperatures mirror those of Badlands National Park (*Graham & Gingerich, 2013*). Bison on Pine Ridge Reservation are hunted throughout the year.

Yellow Bear Pasture (39 km$^2$) is the site of one of the hunted bison populations and is located approximately 30 km southeast of Kyle, South Dakota (43.4250°N, 102.1765°W) and 85 km southeast of Badlands National Park. The landscape is open and undulating and dominated by cheat grass (*Bromus tectorum*), western wheatgrass, buffalograss (*Bouteloua dactyloides*), and little bluestem (*Schizachyrium scoparium*). Canyons also are present, and canyon ridge-tops are dominated by ponderosa pine (*Pinus ponderosa*). Other common tree species include Rocky Mountain juniper, green ash (*Fraxinus pennsylvanica*), and eastern cottonwood (*Populus deltoides*; *Graham & Gingerich, 2013*).

Slim Buttes Pasture (49.7 km$^2$) contains the second hunted bison population on Pine Ridge and is located approximately 20 km northwest of Pine Ridge, South Dakota (43.0255°N, 102.5563°W) and 155 km southwest of Badlands National Park. The same grasses that are found in Yellow Bear Pasture also are present in Slim Buttes. Eastern cottonwood and Rocky Mountain juniper are the dominant tree species (*Graham & Gingerich, 2013*).

## Hunting events

Hunts were conducted year-round in Slim Buttes Pasture and Yellow Bear Pasture with .270 or .25–06 caliber centrefire rifles. In each bison pasture there were three hunting events per month (T Ecoffey, United States Department of Agriculture, pers. comm., 2014). The number of animals taken and the duration of hunts varied. During ceremonial hunts, shooters usually approached bison in motor vehicles (2–3 vehicles with 4–5 people total) to within 50–100 m. Non-lactating cows and 2–3 year-old bulls were usually selected during hunts, though older bulls (greater than five years old) were also removed. During these short ceremonial hunts only one or two bison were taken, and most hunts lasted approximately two hours (*Graham & Gingerich, 2013*).

During non-ceremonial, commercial hunts, up to eight animals were removed in 6 h. Here, a mobile abattoir was utilized, whereby animals were first driven into a holding

 

pasture where two people approached bison to within 25–50 m before discharging their rifles. One animal at a time was downed and carcasses were removed from the pasture and processed at which time shooters re-entered the pasture and repeated the procedure until eight bison had been removed. Remaining animals were then re-released into the pasture (T Ecoffey, United States Department of Agriculture, pers. comm., 2014).

## Data collection

We defined bellowing rate as the total number of distinct bellows by focal bulls that were recorded by the observer during 15-min observations. Bellows were quantified using a stopwatch and a manual hand counter. From a stationary vehicle, we conducted 245, 15-min focal observations (*Altmann, 1974*) of mature bulls between dawn and dusk from 1 July to 13 August 2013 using Vortex binoculars and Leica spotting scopes. We conducted 62 observations among 17 groups in Badlands National Park, 101 observations among 34 groups in Yellow Bear pasture, and 79 observations among 25 groups in Slim Buttes pasture. Mature bulls were defined as bulls greater than four years old (when marked) or those accompanying a cow during the rut when not marked. Group size and composition (i.e., number of attendant males (mature bulls), adult females, juvenile males/females) was defined by the number of animals within 50 m of the focal bull. All mature bulls, whether bellowing or not, that were accompanying a cow were considered to be participating in the rut, and therefore could be included as focal animals. We minimized pseudo-replication by identifying animals based on tag numbers and natural markings. When bulls could not be distinguished by natural markings or ear tags, we visited different groups for each observation and focal individuals were selected haphazardly. We observed bison as close as we could safely approach them throughout the day (50 m), while attempting to avoid oversampling during any particular time of day. Bison did not appear to be bothered by the presence of stationary vehicles in any of the sites as cows and calves routinely passed by parked vehicles within 5 m. Bulls also passed by parked vehicles within 3–5 m while bellowing and fighting other bulls as well as accompanying cows. Therefore, we do not believe that the data were influenced by the presence of observers in motor vehicles. In order to standardize the time period that we observed the rut among pastures, we limited our observations to three weeks after the date of initially observing bellows by bulls in each study site. For example, bellows were first recorded on 6 July in Badlands National Park, 16 July in Yellow Bear pasture, and 17 July in Slim Buttes pasture. Therefore, data were collected in Badlands National Park until 2 August. In Yellow Bear and Slim Buttes pastures data were collected until 9 and 10 August, respectively. Consequently, observation day (described below in the analysis) indicated the progress of the rut (i.e., the number of days that had elapsed after the day that the first bellow was observed).

We characterized adult sex ratios and numbers of bison in each population (Badlands National Park: Population Size = 1,204, Density = 6.2/km$^2$, Adult Male:Female Ratio = 1:10; Yellow Bear Pasture: Population Size = 464, Density = 11.9, Adult Male:Female Ratio = 1:22; Slim Buttes Pasture: Population Size = 184, Density = 3.7, Adult Male:Female Ratio = 1:15) utilizing data collected by Badlands National Park and Oglala Sioux Parks

and Recreation Authority (OSPRA) personnel during fall roundups. Each of the three study sites were considered independent as all pastures were surrounded by fences.

## Ethics Statement

Sensitization can be considered an important indirect effect of hunting; therefore, we observed animals only in areas where bison were subjected to regular human visitation. Data collection did not involve restricted habitat or interference with other species, and was in compliance with institutional (Hofstra University IACUC # 13/14-5) and national guidelines for ethical conduct in the care and use of nonhuman animals in research. We obtained permission to conduct fieldwork in Badlands National Park (Permit #: BADL-2013-SCI-0009) and Pine Ridge Reservation.

## Data analysis

Initially, a series of descriptive statistics of the data were generated. Due to high skew (3.7) and kurtosis (22.2) of bellowing rate, a series of attempted transformations, including square root, log, and numerous others, along with the application of the Johnson family of transformations were run. However, all failed to substantially improve the normality of the data. Therefore, we utilized non-parametric bivariate statistics along with Poisson regression, to appropriately model the distribution of bellowing rate. Additionally, scatterplots of bellowing rate and the continuous predictors (number of attendant males, number of adult females, group size, observation day, and hunting) were investigated in order to ensure linearity as well as the absence of outliers. Linearity was indicated in all cases, with a single extreme outlier found with respect to bellowing rate. However, the removal of this outlier failed to substantially change the results of all analyses conducted; therefore, no cases were removed from the analyses.

Spearman's correlations were initially performed in order to determine the strength, significance, and direction of the relationship between bellowing rate and the predictor variables. An $|r|$ greater than 0.75 was used as the threshold for indicating serious multicollinearity (*Mun, 2008*). We used a Mann–Whitney $U$ test to compare bellowing rate between the two hunted and one protected population, while a Kruskal–Wallis ANOVA with multiple comparisons was used to assess any differences in bellowing rate among the three populations. Finally, two Poisson regressions were conducted in order to determine the extent to which the predictor variables impacted bellowing rate. These multivariate regressions served to determine the impact of each predictor on the outcome of bellowing rate, while holding all other predictors in the model constant. The initial model only included main effects. The second model included an interaction between the number of attendant males and hunting. All analyses were performed using Stata 13.1.

## RESULTS

The Kruskal–Wallis ANOVA comparing the three associated medians was statistically significant ($\chi^2(2) = 81.4$, $p < .001$). Median bellowing rate was an order of magnitude higher in the non-hunted Badlands National Park population (Med = 39, 25th and 75th quartiles = [4, 80], Range = 76) than in the hunted Pine Ridge populations (Med = 0,

**Table 1 Spearman's Rank Correlation.** Spearman's correlations between bellowing rate and possible predictor variables of Plains bison in South Dakota, 1 July to 13 August, 2013.

| Variables | Bellows | No. attendant males | Group size | Day | Hunting |
|---|---|---|---|---|---|
| Bellows | | | | | |
| No. attendant males | −.078 | | | | |
| Group size | .257*** | .177** | | | |
| Day | −.235*** | .573*** | −.240*** | | |
| Hunting | −.576*** | .475*** | −.218*** | .671*** | |
| No. adult females | −.43 | .442*** | .676*** | .099 | .210** |

Notes.
$^*p < .05$, $^{**}p < .01$, $^{***}p < .001$; $df = 242$ in all cases.

25th and 75th quartiles $= [0, 0]$, Range $= 0$). While the median difference between the protected and hunted populations was significant ($z = 8.975, p < .001$), median bellowing rate in Yellow Bear (Med $= 0$, 25th and 75th quartiles $= [0, 0]$, Range $= 0$), and Slim Buttes pastures (Med $= 0$, 25th and 75th quartiles $= [0, 0]$, Range $= 0$) on Pine Ridge was similar ($p > 0.05$).

All Spearman's correlations between bellowing rate and the predictor variables were statistically significant except those between bellowing rate and the number of attendant males and adult females (Table 1). Group size was positively correlated with bellowing rate, while observation day and hunting were negatively correlated with bellowing rate. Correlations between the number of attendant males and the remaining predictor variables (group size, observation day, hunting and number of adult females) were also significant and positive. Group size was significantly and positively correlated with the number of adult females, yet group size exhibited a significant, negative correlation with observation day and hunting.

The results of the first Poisson regression (Table 2) indicate that all predictor variables significantly influenced bellowing rate. Number of attendant males, number of adult females, group size, and observation day (i.e., progress of the rut) positively impacted bellowing rate, while hunting suppressed bellowing rate. For example, the incidence rate ratios (IRR) indicate that average bellowing rate in the hunted populations was about 5% that of the protected population. In contrast, average bellowing rate was predicted to increase by 5% with the addition of each attendant male. The variance inflation factors, taken from an identically specified linear regression model, indicated no substantial multicollinearity using the common cutoff value of 5 (*Oyana & Margai, 2015*).

The second Poisson regression analysis (Table 3) incorporated the interaction between hunting and number of attendant males. All main effects were again statistically significant as was the interaction between hunting and the number of attendant males. The positive interaction between hunting and number of attendant males indicates that bellowing rate increased with the number of attendant males to a greater extent in the protected than in the hunted populations. For example, when the number of attendant males in the protected population was below the mean (4.3 males), the median bellowing rate was 32 bellows/15 min, but it increased to 87 bellows/15 min when the number of attendant males was greater than 4.3. In contrast, median bellowing rate was 0 bellows/15 min both
**Table 2  Poisson Regression.** Poisson regression analysis for bellowing rate of bison bulls from two hunted populations and one non-hunted population in South Dakota, from 1 July to 13 August 2013.

| Variable | IRR | SE | z | p | VIF |
|---|---|---|---|---|---|
| Day | 1.055 | 0.003 | 19.36 | <0.001 | 2.41 |
| No. of attendant males | 1.053 | 0.003 | 17.20 | <0.001 | 1.35 |
| No. of adult females | 1.004 | 0.000 | 9.39 | <0.001 | 1.95 |
| Group size | 1.002 | 0.000 | 11.88 | <0.001 | 2.34 |
| Hunting | 0.053 | 0.003 | −58.43 | <0.001 | 2.28 |

Notes.

$N = 243$, LR $\chi^2(5) = 7085.60$, $p < .0001$; Pseudo $R^2 = .5259$. Data shown are Incident Rate Ratios, standard errors, $z$-statistics, $p$-values, and Variance Inflation Factors (VIF) (derived from an identically specified linear regression analysis).

**Table 3  Poisson regression with interaction.** Poisson regression analysis of the interaction between hunting and number of attendant males on bellowing rate of bison bulls from two hunted populations and one non-hunted population in South Dakota, from 1 July to 13 August 2013.

| Variables | IRR | SE | z | p | VIF |
|---|---|---|---|---|---|
| Constant | 1.153 | 0.095 | 1.73 | 0.083 | |
| No. of attendant males (Std.) | 1.614 | 0.029 | 27.07 | <0.001 | 1.49 |
| Day | 1.060 | 0.003 | 21.50 | <0.001 | 2.41 |
| No. of adult females | 1.004 | 0.000 | 9.68 | <0.001 | 1.95 |
| Group size | 1.002 | 0.000 | 11.79 | <0.001 | 2.39 |
| Hunting (Std.) | 0.236 | 0.006 | −57.91 | <0.001 | 2.30 |
| Hunting *no. of attendant males | 1.227 | 0.014 | 18.24 | <.001 | 1.23 |

Notes.

$N = 243$, LR $\chi^2(6) = 7348.93$, $p < 0.0001$; Pseudo $R^2 = 0.5455$. Data shown are Incident Rate Ratios, standard errors, $z$-statistics, $p$-values, and Variance Inflation Factors (VIF) (derived from an identically specified linear regression analysis). Std. indicates that variable in standardized.

above and below 4.3 attendant males in the hunted populations (Fig. 1). No substantial multicollinearity was indicated, though this was reduced substantially by first standardizing the main effects associated with the interaction effect and using these standardized measures in the calculation of the interaction. Both Poisson regression models were statistically significant ($p < 0.001$). The $R$-squared of both models was above 0.50.

## DISCUSSION

Bulls in the hunted populations on Pine Ridge Reservation bellowed an order of magnitude less frequently than bulls in the protected Badlands National Park population; yet bellowing rates between the hunted populations on Pine Ridge were not statistically different. Hunting had a large negative effect on bellowing rate and was the only variable to do so.

It is plausible that the consistent number (three) of hunting events/month on Pine Ridge, combined with pursuit by—and/or forced close proximity to—motor vehicles and humans for 12 to 18 h/month, may have impacted bellowing behavior. Because bison are generally found in the open, bellowing bulls may be targeted by hunters' sooner than non-bellowing bulls of equal size and age class. Therefore, the best strategy in a risky environment in which there is human hunting may be to dampen bellowing rate. Since there are no natural
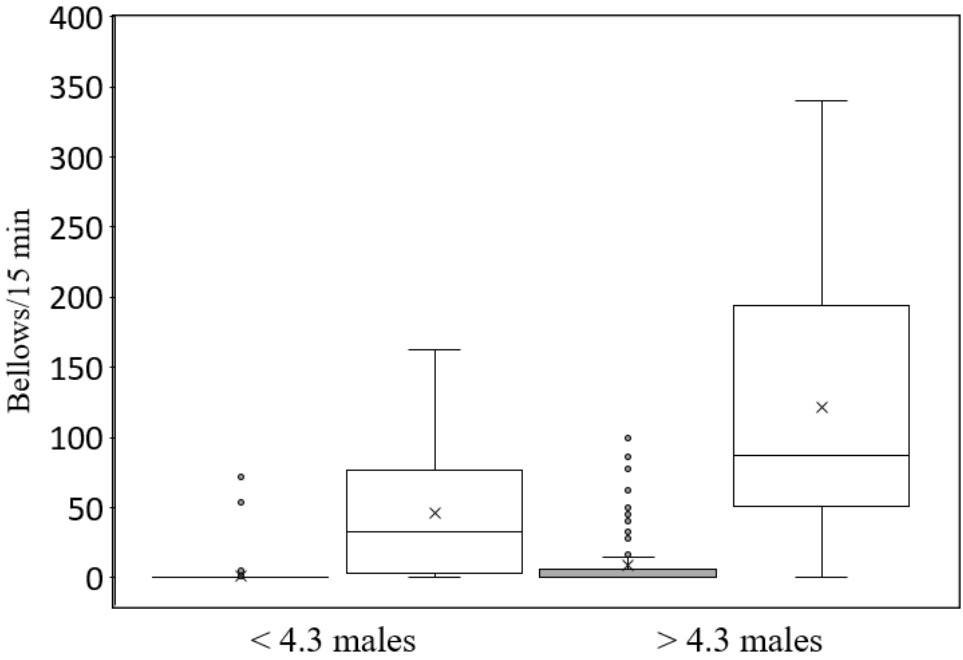

**Figure 1** **Box and whisker plots of median bellowing rate of bison bulls in groups containing fewer than and greater than the mean number (4.3) of attendant males in two hunted populations and one protected population in South Dakota, from 1 July–13 August 2013. Horizontal lines within boxes indicate median bellowing rate. Lower and upper boundaries of the boxes indicate the 25th—and 75th—quartile.** The top and bottom whiskers indicate the maximum and minimum values, respectively, within 1.5 times the interquartile range. Circles show observations more than 1.5 times the interquartile range. The "x" indicates the mean.

predators in any of these populations that are a major threat to bison we can rule out the possibility that decreased bellowing is a response to natural predation.

Approximately 40% of cows 3.5 years and older in Yellow Bear pasture produced calves on a yearly basis (*Oglala Sioux Parks and Recreation Authority, 2013*), while birth rate of similar-aged cows in Badlands National Park was 90% (E Childers, Badlands National Park, pers. comm, 2016). While we do not know the reason for this disparity, it is a possible and disconcerting demographic impact of continual, year-round hunting. Pregnancy rates, however, will also be impacted by the estrus state of cows and there is some concern that copper deficiencies in the diet of bison on Pine Ridge may be impacting reproduction (*Corah & Ives, 1991*, T Ecoffey & C Kelling, 2016, unpublished data). Therefore, fewer females entering estrus would likely have a negative impact on overall bellowing by bulls, because bulls bellow most frequently when in the presence of estrus cows (*Berger & Cunningham, 1994*).

We consider three additional factors that could influence bellowing rate. One, if density were important we would expect the highest bellowing rates to occur (all else being equal) at the highest density, because potential rivals would be physically closer to one another. Bison density in Yellow Bear pasture on Pine Ridge was nearly 2 times higher than that in protected Badlands National Park, yet median bellowing rate of bison in Badlands was an order of magnitude higher than bison in Yellow Bear Pasture. While bison density in

Slim Buttes pasture on Pine Ridge was 3.2 times lower than that in Yellow Bear pasture, median bellowing rate of bulls in both populations was equal. Thus, the influence of density on bellowing rate is not well supported. Two, the operational sex ratio (OSR)—the ratio of reproductively available adult females to adult males—could conceivably influence bellowing rate due to the number of males vying for females. As the operational sex ratio approaches unity (i.e., ratio of males/females approaches 1), we predict increasing bellowing rate due to increasing mate competition among bulls. The operational sex ratio in Badlands National Park (1:9.8) was the closest to unity and bulls in this population bellowed an order of magnitude greater than bulls in either of the Pine Ridge populations. The operational sex ratio in Yellow Bear (1:22) pasture, however, was nearly 1.5 times higher than that in Slim Buttes (1:15.5), yet median bellowing rate in Slim Buttes and Yellow Bear was equal. Thus, the impact of the operational sex ratio on bellowing rate in our study is equivocal and deserves more attention in any future studies. Third, *Berger & Cunningham (1994)* reported that the most important single factor influencing bellow rate was copulatory status of bulls. After males copulated bellow rate dropped to an average of 16% of pre-copulatory bellow rate. Because the Badlands National Park population ($n = 1,205$) was about 2.5 times larger than Yellow Bear ($n = 448$) and Slim Buttes ($n = 463$), it is possible that the number of males that had not copulated and/or new males entering the rut was higher in Badlands than on Pine Ridge, which would have tended to maintain the bellowing rate of bulls at elevated levels in Badlands National Park compared to the hunted populations. Therefore, population size may also influence bellowing rates of bulls during the reproductive period.

Perceived predation risk (*Frid & Dill, 2002*) and disturbance (*Gutzwiller et al., 1994*) diverts time and energy from mating displays in a number of species. Male Tungara frogs (*Engystomops pustulosus*) stifled mating calls and sometimes abandoned mate advertisement in response to perceived predation (*Ryan, 1985*). Male Great snipes (*Gallinago media*) abandoned leks when disturbed or perceived risk to predation (*Fiske & Kålås, 1995*). Among ungulates, overall reproductive success of mule deer (*Odocoileus hemionus*) and caribou (*Rangifer tarandus*) decreased when they were disturbed by ATV's and low-flying aircraft (*Yarmoloy, Bayer & Geist, 1988*; *Harrington & Veitch, 1992*).

All remaining explanatory variables predicted increased bellowing rate. The addition of each attendant male predicted an increase in bellowing rate by approximately 5%. *Berger & Cunningham (1994)* also reported a positive relationship between the number of attendant males and bellowing rate. Although bellowing rate increased as the number of attendant males increased, it did not occur with equal magnitude in the hunted (Pine Ridge) populations vs the protected (Badlands National Park) population. When there were greater than 4.3 attendant males (mean number of attendant males/group), bellowing rate by bulls in the hunted populations was 14 times less than that of bulls in the non-hunted population. When there were less than 4.3 attendant males, bellowing rate by bulls in the hunted populations increased less by a factor of 29 compared with bulls in the non-hunted population. Bulls in the two hunted populations clearly exhibited dampened bellowing rate, and the effect was more pronounced when there were few attendant males.

The passage of each observation day was predicted to increase bellowing rate by 5.5%. We observed the initial three weeks of the rut, and, as a result, infer that cows continually

entered into estrus as the rut proceeded because the estrus state of cows influenced bellowing rate of bulls (and the majority of copulations in the Badlands ecosystem occurred in July (*Berger & Cunningham, 1994*).Group size and the number of adult females in groups also positively influenced bellowing rates, but to a much lesser extent, and their impact can be considered negligible due to low IRR ratios in the regression analyses (Tables 2 and 3).

## Implications for conservation

Human hunting can influence reproductive behavior of ungulates by skewing the sex ratio and/or by removing certain individuals from a population (*Bonenfant et al., 2004*; *Mysterud et al., 2003*). Such changes can profoundly impact mating dynamics (*Mysterud, Coulson & Stenseth, 2002*). Consequently, knowledge of how and why individuals may vary in their reproductive behavior under specific regimes of human hunting will be important for effective management and conservation. It is possible that bison in Yellow Bear and Slim Buttes pastures on Pine Ridge have become sensitized to continual, year-round hunting. Therefore, our results should be of particular conservation relevance to bison because bellows function as an intrasexual display to deter rival males during the mating season (*Berger & Cunningham, 1994*). Bellow amplitude and quality also have been directly associated with mating success (*Wyman et al., 2006*). Because vocalizations of male ungulates appear to impact mating success in a variety of ways, any changes in behavior could alter critical population-genetic parameters like effective population size (*Wright, 1938*) and the capacity for populations to exhibit adaptive evolution. Because mate competition is an important evolutionary process, the IUCN Bison Specialist Group emphasizes the importance that bison express the full complement of natural mating behaviors (*Gates et al., 2010*) and stresses consideration of possible genetic consequences of all management actions, especially those for smaller herds; bellowing behavior is one such component that should be considered. One could argue, however, that bison still express natural mating behavior even if they bellow at diminished rates. It would be risky to assume, however, that decreases in bellowing would be linear among all animals. Differences in age and physical condition likely impact stamina, and therefore, the propensity of males to bellow. Human hunting may exacerbate these differences.

Changes in mating behavior (e.g., propensity to advertise for mates, intimidate, and/or fight rival males) can alter breeding opportunities—or possibly even effectively remove certain individuals from the breeding pool. Population-wide responses to hunting, like fear, are probably transmitted more quickly in social species (*Cromsigt et al., 2013*) like bison, where many individuals (on occasion) can witness conspecifics being downed by humans at close range. Therefore, we suggest that researchers and managers consider the potential that continual, year-round human hunting has to induce behavioral changes in survivors of exploited populations, especially those behaviors that are linked to mating success via mate advertisement and acquisition. In addition to replicating field observations in hunted vs. non-hunted populations, field experiments could also be conducted. For example, bellowing bulls from hunted and non-hunted populations could be recorded, bellows could be categorized, and playback calls could be presented to bulls and cows in order to measure their reaction. A more experimental approach would involve a BACI

design (*Green, 1979*), whereby one non-hunted population would be separated into two equal populations. Hunting could be introduced into one of the populations with the goal to observe any changes in bellowing behavior (including amplitude, rate, pitch, and timbre) of bulls during the rut. Ultimately, if reduced bellowing as a consequence of human hunting is pervasive, then wildlife managers may need to adjust hunting rate and duration, timing (season), and the time lag between hunting events in order to insure that bison are able to express their full repertoire of natural mating behaviors.

## ACKNOWLEDGEMENTS

We thank the administration at Oglala Lakota College (OLC), Oglala Sioux Parks and Recreation Association, OSPRA, and Badlands National Park for permission to conduct this study (Permit #: BADL-2013-SCI-0009). M. Thompson provided invaluable assistance in the field and insights about bison on the Pine Ridge Reservation.

### Funding

This study was funded by NIFA-USDA grant number 2011-38424-30914. Travel funds were made available from a Faculty Development and Research Grant from Oglala Lakota College. The funders had no role in study design, data collection and analysis, decision to publish, or preparation of the manuscript.

### Grant Disclosures

The following grant information was disclosed by the authors:
NIFA-USDA: 2011-38424-30914.
Oglala Lakota College.

### Competing Interests

The authors declare there are no competing interests.

### Author Contributions

- Ronald J. Sarno conceived and designed the experiments, performed the experiments, analyzed the data, contributed reagents/materials/analysis tools, wrote the paper, prepared figures and/or tables, reviewed drafts of the paper.
- Melissa M. Grigione performed the experiments, contributed reagents/materials/analysis tools, reviewed drafts of the paper.
- Alessandra Higa conceived and designed the experiments, contributed reagents/materials/analysis tools, reviewed drafts of the paper, obtained permission to work on Pine Reidge Reservation.
- Eddie Childers contributed reagents/materials/analysis tools, reviewed drafts of the paper, facilitated fieldwork in BadlandsNational Park.
- Trudy Ecoffey contributed reagents/materials/analysis tools, reviewed drafts of the paper, provided one field assistant for me.

### Animal Ethics

The following information was supplied relating to ethical approvals (i.e., approving body and any reference numbers):

Hofstra University IACUC # 13/14-5.

### Field Study Permissions

The following information was supplied relating to field study approvals (i.e., approving body and any reference numbers):

Badlands National Park

BADL-2013-SCI-0009

### Data Availability

The raw data have been supplied as Data S1.

### Supplemental Information

Supplemental information for this article can be found online at http://dx.doi.org/10.7717/peerj.3153#supplemental-information.

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
