# Peer review of "The association between continual, year-round hunting and bellowing rate of bison bulls during the rut"

_PeerJ, doi:10.7717/peerj.3153_

## Round 0.1 · original submission · Minor Revisions

This study examined bellowing rates of bison bulls in two hunted and one protected population. Bellowing rates were much higher in the protected than in the hunted populations. Multivariate analyses indicated that bellowing increased with the number of males, number of females, and group size and that the increase with number of males was greater in the protected population. The authors suggest that the lower bellowing is a response to increased risk of mortality or to the stress associated with the hunt and that hunting may have implications for sexual selection via its effects on reproductive signaling.

Both reviewers found your study publishable but had suggestions for improvements. Reviewer 1 provided additional references in her comments on the pdf, indicated that readers needed more information on the study design including demographic characteristics of the sampled populations, and suggested a more cautious conclusion due to other variables potentially confounding the effect of hunting. Reviewer 2 would like to see the Introduction elaborate possible processes by which hunting could reduce bellowing rates. (This could also be raised briefly in the Introduction and elaborated in the Discussion.) This reviewer would also like to see a more explicit argument concerning the potential confounding effect of number of attendant males on differences in bellowing. (In addition to referring to relevant aspects of your analyses, consider indicating median and ranges for attendant males in the three study populations.) Finally, this reviewer would like you to consider whether lower estrus rates in the hunted populations, potentially but not necessarily related to hunting, could have contributed to the apparent effect of hunting.

I agree with the reviewers and have some suggestions that reinforce and add to theirs. The substantive comments are listed below. These should be treated as a third review rather than an editorial decision i.e., make appropriate changes where my comments are correct and indicate why you disagree for the others. Minor grammatical suggestions and proposals for improved clarity are indicated by highlights and inserted comments on the annotated pdf. In your response, you do not need to comment to each suggestion of my suggestions on the pdf unless you disagree. However, you should copy the suggestions provided by Reviewer 1 from the pdf and include them in your response document.

Editor's comments
Abstract
L23. For readers of the Abstract, I think that a more relevant descriptor such as 'number of days from the start of rut' would be clearer. (I have some questions about this variable; see below.)
L27-29. Revise sentence to avoid causal implication of 'impact'.
L32. The implications are greater than expressing their natural repertoire. You implied that sexual selection may be affected.

Introduction
L52-54. Parentheses and brackets are confusing. Refs needed for locally perceived species.
L72. Increased heart rate and cortisol are not examples of increased vigilance. I suggest 'may increase vigilance or stress levels'.
L105. Also, define a specific objective related to your analysis of covariates.

Methods
L109 and elsewhere. I am not sure what the numbers in parentheses are. I assume they represent latitude and longitude, but the format is not familiar to me. I don't understand the hyperlink on the similar number for Pine Ridge.
L109. Use consistent precision for the area of the 3 study sites. I don't think that precision to hundredths of a square km is needed.
L112ff. From here on, tenses are somewhat inconsistent. It is appropriate to use present to describe conditions of the study site and past to describe your procedures, but check carefully to be sure that this approach is consistent.
L125. For non-US readers, perhaps another couple of words to explain that this area is First Nations tribal land would add clarity, perhaps even adding the name of the tribe and that bison can be hunted throughout the year within their jurisdiction.
L145. Can you say anything more about the frequency? Is the minimum a relevant value here? Note that the Discussion (L270) refers to this as a consistent number which refers to variability rather than minimum values.
L155-157. This is a bit unclear because you do not specify the number of bison killed at each shooting event. The text implies that multiple individuals could be killed rather than just one at a time.
L160-161. This definition implies total bellows heard, not bellows from a focal individual as suggested by L162.
L160-164. If you observed only from vehicles as suggested by L171 but potentially contradicted by L168-169, specify that somewhere in the these first few sentences describing your methods.
L164. Specify that all individuals, whatever their age and sex were included in group size if this was the case. It makes sense to define your other variables here rather than on L179 (where cows are still not explicitly mentioned and group composition is not defined).
L168. This implies that you sampled only one individual per group when individuals were unmarked. Is that correct? Would this result in a different sampling regime on the Reserve and in the Park? If you had a protocol to avoid bias in selecting individuals for focal observations, please describe that protocol. If you didn't have such a protocol, you should specify that focal individuals were selected 'haphazardly'.
L172. Does the description of ignoring vehicles apply to all three areas?
L174-178. The description of observation days is a bit ambiguous. Was there a single day 1 that applied to all sites or separate starts for each site? How did you sample to determine the start of bellowing in each area? The raw data imply that observation day for all areas was the same because there is no separate calculation for each area. However, the study period was about 6 weeks long, so it is not clear how you restricted your study to 3 weeks. The re-stated method in the Discussion (L299-300) does not match what I would have inferred from the description and raw data, so clarification does seem to be in order. A difference among sites could be starting date if rut started synchronously but observations did not (possible topic for Discussion).
L180-182. It would make sense to include the information on the population size, density, and sex ratio in the site descriptions.
L190. Should you say something about the permission/permit process for observations on the Reserve?
L199. Don't repeat the definition of observation day if clearly defined above.
L200. Perhaps it would be appropriate here to indicate how you defined the hunting variable. I assume that it was presence/absence, but you did not specify how you defined the numerical values. Also, it is not clear whether examining linearity as indicated here or Spearman correlations as specified below is valid with only presence absence data. Once clearly defined, don't repeat yes/no on L211.
L210-211. If predictor variables are clearly defined above, they don't need to be repeated here.

Results
L217ff. The presentation of your main results needs some clarification.
• The symbol used for the mean is not standard. Perhaps Adobe changed the symbol for x bar when you made the pdf.
• However, I don't think you should be using the mean, anyway. For non-normal data, descriptive statistics should be non-parametric, so the appropriate measures would be the median, a measure of variation such as the 25th and 75th quartiles, and the range.
• In addition, you should give the units for the values (bellows/15 min?)
• An alternative order for information may be more logical and concise. Consider starting with the Kruskal-Wallis to show that the 3 sites differ. Then, write something along the lines of 'Bellowing rate in the National Park (values) was significantly higher than that in both the Yellow Bear (values; t, df, p) and Slim Buttes (values: t, df, p) pastures in the Reserve. However, Yellow Bear and Slim Buttes did not differ (t, df, p).' I don't see that the means or stats for combined values in the Reserve adds to this comparison. If it is important, however, you also need to indicate whether combined values is based on a mean of all values or a mean of the two means, if sample sizes differ between sites.
• Note that in the above suggested description, the test statistic, df and p should all be provided.
• Considering the concerns of the reviewers and my own reading, I think that a table of the descriptive data for the other variables would be of value here. For each site, each variable would have median, quartiles and range. The dates would also be given if observation day calculation differs among sites.
• Referring to this table would allow you to indicate in the Results any major differences between sites in your measured variables that might confound the effect of hunting as well as stating which variables are similar between sites and to develop the implications in the Discussion.
L228. It seems odd to mention only the correlations that are non-significant. Mention the significant ones as well, also indicating whether they are positive or negative.
L230. Again, direction of effect.
L231. Revise to clearly indicate what is correlated with what; three variables mentioned make the relationships ambiguous.
L231-232. Presumably the correlation between day and hunting is a reflection of your study design, not an observed pattern. If you clarify the relationship between site and study period in the Methods, you probably don't need to mention it here.
Table 1. I agree with the reviewer that the table would be easier to read if you added the variables to the column heads instead of using numbers. (Remove the numbers that appear to be footnotes from the row headings.) It will also be easier to read if you align the decimals.
Tables 2 and 3. Because # is not a standard international abbreviation for number, I suggest writing 'no. of' instead. If you follow suggestions on the pdf for changing Table 2, apply the same to Table 3. I don't see the logic of your order of variables. They don't seem to match the previous order of presentation or the values of any column. Zeros are needed before the decimal, except for p-values if you prefer not to use them there. 'Std' is not defined in the footnote for Table 3. I was surprised that standardizing a dichotomous variable (hunting) makes a difference. I could easily be wrong about this, but you could check with a statistician if there is any doubt.
L244-253. I found this section a bit confusing to understand the first time through.
• If number of attendant males are not normally distributed, it would be better to divide the data at the median rather than the mean.
• The interaction does not really seem to be the cause of dichotomization. I think you can just say that you dichotomized.
• The text describing the data is not very clear and is incorrect in stating that the difference was twice as large with fewer attendant males. The difference is 44 with few bulls and 113 with many bulls. It is the ratio that is higher with few bulls. This opposite trend for difference and ratio seems to be the source of the confusion. For this reason, the repetition emphasizing the ratio in the Discussion (L297) is also misleading and not reflective of the interaction measure.
• I think a figure showing bar graphs to represent the mean bellowing rate (including units) for low and high attendance in hunted and protected populations would help you make your case more cogently.
• Check for consistency in using non-hunted vs. protected throughout the text.
• I suggest writing out greater than and less than instead of using the signs. This seems potentially awkward in phrases such as '> twice as large'.
• Consider whether a text along these lines seems clearer and more concise: 'The positive sign of the hunting*no. of attendant males interaction means that bellowing rate increases with number of attendant males more in the protected than in the hunted populations. Dividing the data into observations at or below the median bellowing rate (X bellows/15 min) and those above the median show that hunted populations bellow about 44 bellows/15 min less than protected populations (29 times less) when the number of attendant males is low. They bellow about 113 bellows/15 min less than protected populations (14 times less) when the number of attendant males is high (Fig. 1).'

Discussion
The Discussion needs stronger organization and less redundancy. Much of it is quite repetitious of the Introduction and Results. It is appropriate to briefly remind readers of key results as your discussion develops but an extended repetition is not needed.
• For the effect of hunting, a more self-critical approach would be appropriate, considering possible confounding unmeasured variables in the Reserve and the Park. This is where the need for additional replication, clearly stated in the Abstract could be emphasized. It is also where the literature review of effects of predation risk on vocal sexual signaling would go, focusing on the relationships between vocalizations and risk. At present, the reference to reducing fitness-enhancing behavior in the topic sentence (L279) is distracting because it is more relevant to the later discussion of implications. This would be the logical place to state as your claim to the first evidence for an effect of hunting on vocalizations and a consideration of possible mechanisms by which hunting might affect vocalizations.
• This would be followed by paragraphs with similar structure (perhaps excluding discussion of possible mechanisms), but potentially briefer, addressing the effect of your other variables. Rather than repeating your results for the effect of attendant males in such detail, it might be more interesting to examine the quantitative findings to see if the Park or Reserve more closely matches Berger and Cunningham.
• Your final section addresses implications of the reduction in bellowing in the hunted populations. However, the focus is lost a bit by a topic sentence in the first sentence that returns to impact of hunting on behavior rather than the implication of behavioral changes for conservation. The next paragraph is more on-target, but provides detail on the population genetic implications raised without as much explanation in the previous paragraph.
• L263-269. This review of the literature is quite redundant to the Introduction. Normally, the Discussion would develop and extend a review of relevant literature that had been presented more briefly in the Introduction. This also seems more relevant to the discussion of the implications than to the first paragraph of the Discussion.
• L269. The last sentence needs a reference or an indication that this is your own inference from the previous references.
• L295. Misleading description. Bellowing rate did not drop but increased less.
• L303. Specify the evidence for the negligible effect of group size and number of females. Is there any literature related to these variables?
• L310-311. The contrast between mating success and deterring rivals is misplaced because mating success is often decreased by defending against rivals. Do you mean that bellowing affects female mate preference as well?
• L314. Is the recommendation for expressing the full complement of mating behavior based on welfare or ecological/evolutionary concerns? If the bison still bellow, albeit at a reduced rate, are they not still expressing that aspect of their behavior? If all individuals lower their rates, is it possible that relationships among them would be unchanged? Note that the related statement in your Abstract appears to have a much more limited scope than developed in this paragraph.
• L326. It is appropriate to call for more replication, but it is not clear what 'fully replicated' means. Would it be out of place to suggest an experimental approach? It seems that you could slightly elaborate why your hunting management suggestions might work. Indeed, in calling for increased replication even with observational rather than experimental data, it seems that attention to hunting protocols and temporal patterns would be very appropriate.
• L328-331. Your paragraph ends on a point very similar to the end of the previous paragraph, reinforcing the need for stronger organization of the Discussion. A broader, more synthetic final sentence would be desirable.

References.
Please check all references carefully. There are missing titles, caps included in journal article titles, spelling errors, inconsistent and non-standard journal abbreviations, missing italics from species names, missing caps from proper names, pages and editors missing from book chapters.

·

Basic reporting

The causation of bellowing is still poorly understood and can be influenced by a large number of factors. There are some more studies that are worth referring to in order to better understand the possible causation and function of the behaviour (cfr comments in pdf).

Experimental design

Some information is lacking (eg sampling effort, sampling days in each location).

Validity of the findings

Can you report more relevant data such as number of groups encountered in function of sampling effort, group sizes & density & sex ratios.
The conclusion reaches too far in my opinion and remains speculative given the limited sample size (one hunted population; two not-hunted). There may be a number of other potentially determining factors that were or could not be included in the study (eg spatial dynamics, number of new contestants, number of new females,...).It should be stated more clearly that the factors that were included in the study are not the only potential crucial factors influencing bellowing rate.

Additional comments

I have added my comments in the pdf version of the article.

Reviewer 2 ·

Basic reporting

Standards are adhered to.

Experimental design

Design and analysis are simple and straightforward.

Validity of the findings

No comments

Additional comments

The introduction is clear and to the point. Changes in bellowing rate may indeed alter fitness, so the study is well justified. It would have been more satisfying to state why bellowing rate might decline in hunted populations. As now written, the justification for the hypothesis (line 80-81) is based on a personal observation that bellowing rate seemed lower in the hunted populations – it would be better to connect this to a mechanism. For example, does bellowing increase the chances of being targeted when hunting occurs? Or do hunted males move more and thus have less time to bellow? I’m no expert on bison, but surely some reasoning could be applied to the hypothesis.
The only mention of a causal connection between hunting and bellowing rate begins on line 272: “Because bison are generally found in the open, perhaps the best strategy in a risky environment is to remain quiet even during a season when it is advantageous to vocally advertise for mates.” This is not clear – do hunters locate their prey by listening for them? What does being found in the open have to do with increasing or decreasing vocalization?
L 291-297 The effect of attendant males in the two types of populations is striking. I assume hunted populations are smaller or at lower density than non-hunted populations, and I worry that differences in these factors led to the differences in bellowing rate rather than some caution being displayed by males in the hunted population. Did the regressions and investigations of multicollinearity eliminate this concern? If so, please state it clearly.
L 298-302 If the pregnancy rate in hunted populations is only half that of hunted populations (L 274-276), and estrus rates affect bellowing rate, might some of the difference in bellowing rate be due to fewer females being in estrus in hunted populations, rather than some caution being displayed by males in the hunted population? If this is not likely, please state why.
Details:
L 120 The similarity between a minus sign and an en dash leads to some confusion in the mention of temperature range.
L 145, 149, 175 Please change the hyphen in 2-3 to 2–3 (an en dash) to be consistent with similar usage in the paper
L 198 – 200 Missing right parenthesis
L 218 Not clear what ´x is.
L 358, 380, 411, 443 All initial caps – please change to conform to other citations
L 363 No title
L 399 Caps needed

---

## Round 0.2 · accepted · Accept

The manuscript is now ready for publication, except for the minor edits you and I discussed via email and that you can resolve in production.

Figure 1 needs to be replaced with a box plot. Corresponding text L272-282 should be replaced as we discussed. On Table 3, L3 should be 2001, not 201 and # should be replaced with 'no. of' in two places.